# Inhibition of p300/CBP-Associated Factor Attenuates Renal Tubulointerstitial Fibrosis through Modulation of NF-kB and Nrf2

**DOI:** 10.3390/ijms20071554

**Published:** 2019-03-28

**Authors:** Sungjin Chung, Soojeong Kim, Mina Son, Minyoung Kim, Eun Sil Koh, Seok Joon Shin, Cheol Whee Park, Ho-Shik Kim

**Affiliations:** 1Department of Internal Medicine, College of Medicine, The Catholic University of Korea, 222, Banpo-daero, Seocho-gu, Seoul 06591, Korea; chungs@catholic.ac.kr (S.C.); sma0917@naver.com (M.S.); gomy33@naver.com (M.K.); fiji79@catholic.ac.kr (E.S.K.); imkidney@catholic.ac.kr (S.J.S.); cheolwhee@hanmail.net (C.W.P.); 2Department of Biochemistry, College of Medicine, The Catholic University of Korea, 222, Banpo-daero, Seocho-gu, Seoul 06591, Korea; soojeong107@hanmail.net

**Keywords:** histone acetyltransferase, p300/CBP-associated factor, kidney fibrosis, inflammation, oxidative stress, apoptosis

## Abstract

p300/CBP-associated factor (PCAF), a histone acetyltransferase, is involved in many cellular processes such as differentiation, proliferation, apoptosis, and reaction to cell damage by modulating the activities of several genes and proteins through the acetylation of either the histones or transcription factors. Here, we examined a pathogenic role of PCAF and its potential as a novel therapeutic target in the progression of renal tubulointerstitial fibrosis induced by non-diabetic unilateral ureteral obstruction (UUO) in male C57BL/6 mice. Administration of garcinol, a PCAF inhibitor, reversed a UUO-induced increase in the renal expression of total PCAF and histone 3 lysine 9 acetylation and reduced positive areas of trichrome and α-smooth muscle actin and collagen content. Treatment with garcinol also decreased mRNA levels of transforming growth factor-β, matrix metalloproteinase (MMP)-2, MMP-9, and fibronectin. Furthermore, garcinol suppressed nuclear factor-κB (NF-κB) and pro-inflammatory cytokines such as tumor necrosis factor-α and IL-6, whereas it preserved the nuclear expression of nuclear factor erythroid-derived 2-like factor 2 (Nrf2) and levels of Nrf2-dependent antioxidants including heme oxygense-1, catalase, superoxide dismutase 1, and NAD(P)H:quinone oxidoreductase 1. These results suggest that the inhibition of inordinately enhanced PCAF could mitigate renal fibrosis by redressing aberrant balance between inflammatory signaling and antioxidant response through the modulation of NF-κB and Nrf2.

## 1. Introduction

The DNA double helix in eukaryotic cells is packaged into a compact structure called chromatin with the assistance of two major classes of proteins: histones and non-histones [1]. Among the histones, H1 is known as the linker histone while H2A, H2B, H3, and H4 are considered core histones. Each core histone has a flexible N-terminal tail that consists of amino acids prone to posttranslational modifications including acetylation, methylation, phosphorylation, ubiquitylation, sumoylation, ribosylation, citrullination, deamination, and proline isomerization [1]. These histone modifications play an important role in assembling heterochromatin and maintaining gene boundaries between transcribed and non-transcribed genes. Histone acetylation and deacetylation, orchestrated by histone acetyltransferase (HAT) and histone deacetylases (HDAC), respectively, regulate the opening and closing of the chromatin structure to guide gene expression machinery [2]. Acetylation of histone by HATs results in the relaxation of the chromatin structure, which promotes gene expression, whereas the removal of the acetyl group by HDACs represses gene expression [1,3]. HATs are also known to be responsible for the accessibility of transcription factors and transcriptional activation [4,5]. Physiological equilibrium of histone acetylation is often disturbed by histone hyperacetylation caused by either HAT activity or lack of HDAC activity [1,3]. In some instances, histone hyperacetylation may result in the overexpression of genes unfavorable for cell survival. Previous studies have demonstrated that dysregulation of histone acetylation is linked to several diseases including cancer, neurodegenerative disease, and chronic inflammation [6,7].

HATs can be grouped into four families according to sequence homology and structural features as well as functions [1]: the Gcn5-related N-acetyltransferase (GNAT) family, MYST (MOZ, Ybf2, Sas2, and Tip60) family, p300/CBP family, and nuclear coactivators (NRC) family. HATs belonging to the GNAT family are associated with the acetylation of lysine residues on histones H2B, H3, and H4 [1]. As a member of the GNAT family, p300/CBP-associated factor (PCAF) is involved in the regulation of cell transcription, progression, and differentiation [1,8]. For PCAF activity, lysines (K) 9 and 14 of histone H3 are preferred substrates [8]. PCAF can also bridge transcriptional factors to the transcriptional complex to provide appropriate levels of gene activities in cells in response to extracellular stimuli [9]. To date, some studies have suggested that PCAF can regulate the expression of inflammatory molecules [1,5,9,10]. PCAF has been reported as one of the genes significantly regulated by activated protein C in human macrophages [4]. Importantly, PCAF is considered to play a role in microglial inflammation through acetylation-dependent nuclear factor kappa B (NF-κB) activation [7].

Renal fibrosis involves a complex multistage process that is orchestrated by a network of cytokines, chemokines, growth factors, adhesion molecules, and signaling processes [10]. A few studies have shown that PCAF is associated with increased apoptosis and upregulation of some inflammatory genes such as intercellular adhesion molecule-1, vascular cell adhesion molecule-1, and monocyte chemotactic protein-1 as well as NF-κB in the kidneys of streptozotocin-induced diabetes rats, *db*/*db* mice, and lipopolysaccharide (LPS)-injected mice [2,11]. Oxidative stress is also implicated in the pathogenesis of various forms of renal injury [12,13]. However, little is known about the involvement of PCAF in oxidative stress in the pathogenesis of renal fibrosis. Therefore, the objective of this study was to evaluate whether PCAF might be involved in the regulation of epithelial-mesenchymal transition (EMT), oxidative stress, and inflammatory molecules in the kidneys of nondiabetic mice with unilateral ureteral obstruction (UUO). In addition, mechanisms of renal protection by garcinol, a PCAF inhibitor, were investigated in this study. Identifying additional roles of PCAF and epigenetic effects on renal fibrosis may further increase the possibility of developing therapeutic strategies against renal diseases.

## 2. Results

### 2.1. Expression Levels of PCAF and Histone Acetylation in Kidneys Are Increased after UUO

We initially tested whether UUO surgery could cause changes in PCAF expression in obstructed kidneys. Immunohistochemistry analysis and immunoblotting showed that PCAF staining in kidneys was markedly increased on both days 3 and 7 after UUO when compared to that in the kidneys of sham-operated mice (Figure 1A–D). Such elevation in PCAF expression was reduced by the PCAF inhibitor garcinol. Consistent with results of PCAF expression, acetylation of K9 on histone H3 (H3K9ac) was also increased in obstructed kidneys. Such an increase was attenuated by garcinol treatment, where the finding was remarkable at day 7 after UUO (Figure 1E,F).

### 2.2. Inhibition of PCAF Leads to Attenuation of Fibrotic Changes in Obstructed Kidneys

Based on Masson’s trichrome staining, renal tubulointerstitial fibrosis was increased after UUO when compared with that of the sham-operated mice. However, garcinol treatment significantly reduced fibrotic lesions after obstructive injury (Figure 2A,B). Total collagen content and type IV collagen (Col IV) expression in obstructed kidneys were significantly increased after UUO. Both were effectively decreased by garcinol treatment (Figure 2C–E). Garcinol administration also reduced the expression increase of transforming growth factor (TGF)-β1 induced by UUO (Figure 2D,F,G). Collectively, these results indicate that PCAF plays a role in the progression of kidney fibrosis.

### 2.3. Inhibition of PCAF Debilitates Epithelial-Mesenchymal Transition in Obstructed Kidneys

Renal tubular epithelial cells of patients with chronic kidney disease (CKD) undergo an EMT while increased expression of fibroblast-associated proteins and matrix deposition contribute to renal fibrosis [14]. Immunohistochemical staining revealed that levels of α-smooth muscle actin (α-SMA), a specific marker for myofibroblast activation, were significantly decreased in obstructed kidneys on both days 3 and 7 after the administration of garcinol (Figure 3A,B). Results of quantitative real-time polymerase chain reaction (qRT-PCR) revealed that increases in renal mRNA levels of α-SMA, vimentin, and fibronectin were significantly reduced by garcinol treatment at day 3 after UUO (Figure 3C–E). Levels of matrix metalloproteinase (MMP)-2 and MMP-9 enhanced in obstructed kidneys were also decreased with garcinol treatment at both days 3 and 7 after UUO (Figure 3F,G). Garcinol treatment also decreased the mRNA level of E-cadherin on day 7 after UUO (Figure 3H). 

Garcinol decreased the renal mRNA level of vascular endothelial cadherin (VE-cadherin) at day 7 after UUO (Appendix A). It also decreased the levels of endothelial marker CD31 at day 3 post-UUO (Appendix A). Levels of tubular epithelial markers such as fibroblast-specific protein (FSP)-1 and galectin-3 that were elevated after UUO were also decreased at days 3 and 7 after garcinol treatment (Appendix A).

### 2.4. Inhibition of PCAF Reduces Inflammation in Obstructed Kidneys

In UUO mice, F4/80-positive cells infiltrated the tubulointerstitium of the obstructed kidney (Figure 4A,B). After garcinol treatment, the number of F4/80 cells was significantly lower, indicating the inhibition of inflammation. Garcinol administration also significantly attenuated renal mRNA levels of interleukin (IL)-6 at day 7 after UUO and those of tumor necrosis factor (TNF)-α at days 3 and 7 post UUO (Figure 4C,D). 

We then assessed the influence of a PCAF inhibitor on the important inflammatory response pathway NF-κB. Increases in both NF-κB p65 protein and its phosphorylation at Ser-536 were observed on days 3 and 7 after UUO (Figure 4E–G). This activation was significantly suppressed by garcinol treatment. Collectively, these results revealed that NF-κB activation and the subsequent transcription of inflammatory genes after UUO were effectively inhibited by a PCAF inhibitor. This might be a mechanism underlying the anti-inflammatory effect of garcinol on renal fibrosis.

### 2.5. Inhibition of PCAF Decreases Oxidative Stress and Increases Antioxidant Enzymes in Obstructed Kidneys

It has been reported that oxidative stress contributes to the pathogenesis of UUO [13]. Thus, we examined the effect of the blockade of PCAF on renal oxidative stress. NADHPH oxidases of the Nox family are the most prominent source of reactive oxygen species (ROS). The function of these enzymes is ROS generation [15]. Protein levels of Nox2 were markedly increased at both days 3 and 7 after obstructive injury. However, these increases were significantly attenuated by garcinol (Figure 5A,B). 

We also evaluated the expression of antioxidant enzymes. On day 7 after UUO, mice kidneys had suppressed expression of heme oxygenase-1 (HO-1). However, garcinol administration resulted in significantly increased expression of the HO-1 protein (Figure 5A,C). Protein expression levels of catalase and superoxide dismutase 1 (SOD1) decreased by UUO were also significantly restored by the administration of garcinol (Figure 5A,D,E). NAD(P)H:quinone oxidoreductase 1 (NQO1) demonstrated an increase after obstructive injury while garcinol was able to further increase its expression at all timepoints (Figure 5A,F). Given that these enzymes are major target genes of nuclear factor-erythroid-2-related factor 2 (Nrf2) [16], these results suggest that garcinol could affect Nrf2 activation. The expression of nuclear Nrf2 was decreased by UUO but was significantly increased by garcinol treatment (Figure 5A,G). These results indicate that inhibition of PCAF by garcinol could preserve the activation of Nrf2 and subsequent expression of its target genes.

### 2.6. Inhibition of PCAF Decreases Oxidative Stress and Increases Antioxidant Enzymes in Obstructed Kidneys

Considering that apoptosis was increased in the kidneys of mice with UUO, we determined the effect of garcinol on renal apoptosis by performing the terminal deoxynucleotidyl transferase dUTP nick end labeling (TUNEL) assay and Western blot for pro- and anti-apoptotic proteins. The number of TUNEL-positive cells was increased in the kidneys of UUO mice but was markedly decreased by garcinol treatment (Figure 6A,B). UUO caused a decrease in the ratio of Bcl-2 to Bax. This decrease was significantly increased by garcinol (Figure 6C,D).

## 3. Discussion

The current study was designed to explore the role of PCAF in renal injuries induced by UUO. We showed that PCAF was substantially activated in nondiabetic kidneys undergoing renal inflammation and fibrosis. Our results indicate that PCAF might be able to simultaneously influence numerous mediators and signaling pathways of inflammation, oxidative stress, EMT, and apoptosis of diseased kidneys. Regarding the overall effects of PCAF on these processes, previous studies have primarily focused on the role of PCAF in inflammation. 

By acetylating histone H3 or H4, histone acetyltransferases (HATs) such as CBP/p300 and PCAF can change the chromatin structure from heterochromatin to euchromatin, which facilitates the binding of transcriptional factors to promoters or enhancers in chromatin [17]. To be consistent with this hypothesis, CBP and PCAF are required for the transcriptional activity of NF-κB, the central regulator of inflammation [9,18]. Moreover, the HAT activity of PCAF enhances the transcriptional activity of NF-κB [19], suggesting that PCAF may play a critical role in inflammation by modulating the transcriptional activity of NF-κB. In this study, the expression of proinflammatory cytokines including IL-6 and TNF-α, transcriptional target genes of NF-κB, was upregulated associated with PCAF activation and as expected, the inhibition of PCAF activity with the treatment of garcinol reduced the upregulation of pro-inflammatory cytokines, resulting in the prevention of renal fibrosis. Therefore, it can be hypothesized that harmful stimuli in the kidney such as UUO activate PCAF activity as well as the NF-κB signaling pathway, leading to pro-inflammatory gene expression, thus, the inhibition of the HAT activity of PCAF contributes to anti-inflammation, based on our results. Notably, garcinol treatment prevented the activation of NF-κB (Figure 4), suggesting that PCAF might be involved in the activation pathway of NF-κB. Although the mechanism of how PCAF is involved in the activation of NF-κB in UUO-induced renal fibrosis was not investigated in this study, this finding suggests that protein kinases or molecules regulating the activity of NF-κB could be regulated by PCAF, as in the case of a virulence factor of Yersinia, YopJ, acetylating IκB kinase (IKK) [20], which may highlight the feasibility of PCAF as a new therapeutic target in the pro-inflammatory process. 

Another important finding in our study was that PCAF upregulation was associated with increased oxidative stress. Oxidative stress by elevation in ROS plays a pivotal role in renal fibrosis [21]. It is known that the balance between ROS production and the ROS scavenging system is an important homeostatic regulator in the progression of kidney fibrosis [21,22]. ROS produced by NADPH oxidases have been implicated in many physiologic and pathophysiologic processes. It plays an important role in cell signaling as a second messenger. It is known to mediate hormonal effects, regulate ion channel activity, oxygen sensing, adipocyte differentiation, gene expression, reproduction, cell growth, senescence, and apoptosis [22,23]. Furthermore, growing evidence indicates a role of NADPH oxidases in renal fibrosis [22]. In our experiment, elevated levels of renal Nox2 was downregulated in the kidneys of UUO mice after treatment with garcinol. Although no studies have been published on the relationship between PCAF and Nox2, PCAF may play a role in the generation of ROS via Nox2 expression in the kidney. Oxidative stress is not only associated with the excessive production of ROS, but is also associated with disturbances in cellular antioxidant systems [24]. As described in previous findings [16,25], UUO is linked to the reduced activity of antioxidant enzymes including HO-1, SOD1, and catalase. Genes encoding for these antioxidative proteins are transcribed by the activation of the Nrf2-Kelch-like ECH-associated protein 1 (Keap1)-antioxidant response elements (ARE) pathway [26]. Under a normal state, Nrf2 is retained by Keap1 in the cytoplasm [25,26]. With oxidative stress, Nrf2 will dissociate from Keap1 and translocate to the nucleus where it binds to ARE [26]. NF-κB also regulates renal Nrf2 expression in the kidneys of CKD rats by promoting Keap1 and subsequent Nrf2 ubiquitination or interfering interaction of Nrf2 with ARE sequences [27,28]. Nrf2 is believed to be an anti-inflammatory modulator for the regulation of NF-κB by decreasing IκBα phosphorylation, thereby reducing the nuclear accumulation of NF-κB [26,29]. In the present study, we found a decrease in nuclear Nrf2 and an increase in nuclear NF-κB with the upregulation of PCAF in the kidneys of UUO mice. After treatment with a PCAF inhibitor, there was an increase in the accumulation of nuclear Nrf2 and a subsequent increase in antioxidant enzymes as well as decreased accumulation of nuclear NF-κB. These results suggest that PCAF is involved in the regulation of both the NF-κB and Nrf2 pathways and that PCAF inhibition by garcinol may contribute to renoprotection by modulating the balance between inflammatory and antioxidant modulators.

In this study, we also evaluated the possible association between PCAF and EMT-related factors using both in vivo and in vitro experiments. Numerous factors can regulate EMT including growth factors, cytokines, hormones, and extracellular cues of different pathways [14,30]. Despite some debate on the contribution of EMT to renal fibrosis, it has been thought that EMT is an adaptive response of the epithelial following chronic injury that plays an integral role in the development of renal fibrosis [14,31]. During EMT, renal tubular cells lose their epithelial phenotypes evidenced by changes in epithelial markers such as E-cadherin and acquire new characteristic features of mesenchymal cells including increased expression of mesenchymal markers such as vimentin, fibronectin, MMP-2, and MMP-9 [31,32,33,34]. It has been reported that phosphorylation of upstream targets can influence the regulation of EMT-related factors [32,35]. For example, nuclear localization of NF-κB through the phosphorylation of IκB kinase-α and IκB will in turn interact directly with the promoter of some EMT-related factors to regulate them at the transcriptional level [32]. Given that the exposure of cells to garcinol significantly downregulates the expression of NF-κB and inhibits its nuclear translocation [35], one possible mechanism by which PCAF regulates EMT would be through the control of NF-κB. 

The main limitation of this study was the absence of renal functional information. However, many previous studies have reported that BUN or serum creatinine is not significantly affected by UUO because of the presence of a contralateral kidney with good kidney function [36,37], indicating that BUN or serum creatinine is not a good indicator of renal function in an animal model of UUO [25]. In addition, although it would have been best for this study to separate the renal medulla from the renal cortex, all experiments were performed in whole kidneys. It has been reported that glomeruli initially remain viable by remodeling of Bowman’s capsule after UUO and they could become non-functional with proximal tubular damage [38]. Cortical damage in the obstructive kidney was evidenced by a 50% reduction in the volume fraction of proximal tubular mass with unaltered relative mesangial area of glomeruli [39], suggesting that it would be justifiable to use whole kidney homogenates rather than medullary homogenates in this study.

In summary, we demonstrated that PCAF might play a major role in organizing whole fibrogenic processes through the modulation of inflammatory mediators, oxidative stress, and EMT. Garcinol treatment appears to appease excessively activated PCAF in diseased kidneys. It may correct the imbalance between activities of NF-κB and Nrf2, driving decreased transcription of inflammatory mediators, and increased transcription of antioxidants. At the top-level of cellular processes, epigenetic mechanisms in chromatin can regulate gene expression, cellular identity, phenotypic variations, and disease states without altering the underlying DNA sequence [35]. For renal protection, epigenetic control of gene regulation might be a novel therapeutic strategy that could modulate many different pathological pathways simultaneously.

## 4. Materials and Methods

### 4.1. Animals

All experiments were performed in accordance with protocols approved by the Institutional Animal Care and Use Committee of The Catholic University of Korea, Yeouido St. Mary’s Hospital (No. YEO20161601T). Male C57BL/6 mice weighing 20-25 g (OrientBio, Inc., Seoul, Korea) underwent left ureteral ligation with 4-0 silk thread under general anesthesia as described previously [16,29]. The sham operation was performed in a similar manner without ligation. Mice received an intraperitoneal injection of garcinol (Sigma-Aldrich Co., St. Louis, MO, USA) at 0.5 mg/kg/day or vehicle (200 μL) alone at 3 or 7 days after operation. The dose of garcinol chosen in this study was expected to be non-toxic based on previous data [40,41]. Mice were divided into five groups (n = 6 per group): sham-operated control mice (Sh), control mice sacrificed on day 3 after UUO (U3C), garcinol-treated mice sacrificed on day 3 after UUO (U3Ga), control mice sacrificed on day 7 after UUO (U7C), and garcinol-treated mice sacrificed on day 7 after UUO (U7Ga). At the end of the experiment, kidneys were harvested for histological evaluation and molecular analysis.

### 4.2. Histology and Immunohistochemistry

As previously described [16,25], 4% phosphate-buffered paraformaldehyde-fixed kidney section was stained with Masson’s trichrome to evaluate the severity of tubulointerstitial fibrosis. To assess the expression of PCAF (Abcam, Milton, Cambridge, UK) or H3K9ac (Abcam), kidney sections were reacted with the anti-PCAF antibody or anti-H3K9ac antibody followed by incubation with an anti-rabbit antibody (Vector Laboratories, Burlingame, CA, USA). To determine the extent of myofibroblasts or macrophages, sections were treated with anti-α-SMA or F4/80 (all from Abcam). After washing with PBS, all sections were incubated with peroxidase-conjugated anti-mouse IgG (Jackson ImmunoResearch Laboratories, West Grove, PA, USA) as a secondary antibody at room temperature for 90 min and then reacted with a mixture of 0.05% 3,3′-diaminobenzidine containing 0.01% H_2_O_2_ for color reactions. For quantitative assessment, more than 20 fields were randomly selected and analyzed with MetaMorph image analysis software (Molecular Devices, Sunnyvale, CA, USA). To examine the degree of apoptosis, the TUNEL assay (Millipore, Billerica, MA, USA) was performed according to the manufacturer’s protocol. TUNEL-positive cells were evaluated in 20 randomly selected tubulointerstitial fields for each section using ImageJ 1.49 software (National Institutes of Health, Bethesda, MD, USA). All histologic slides were assessed in a blinded manner.

### 4.3. Renal Collagen Content Assay

The total collagen content of kidney tissue was measured by acid hydrolysis of the kidney tissue as described previously [16,25]. Briefly, each kidney sample was hydrolyzed in 6 N HCl for 18 h at 110 °C and then dried at 75 °C. After solubilizing in a citric acid collagen buffer, samples were filtered through centrifugal filter units (EMD Millipore, Darmstadt, Germany) and then oxidized with chloramine-T solution. To start the color reaction, 100 μL of Ehrlich’s reagent (Fisher Scientific, Fair Lawn, NJ, USA) was added. Sample absorbance was measured at 550 nm. Total collagen in the kidney tissue was calculated based on the assumption that collagen contained 12.7% hydroxyproline by weight.

### 4.4. Western Blot Analysis 

Total proteins of kidney tissues were extracted using a PRO-PREP Protein Extraction Kit (iNtRON Biotechnology, Seongnam-si, Gyeonggi-do, Korea). Nuclear proteins of kidney tissues were extracted using a NE-PER Nuclear and Cytoplasmic Extraction Kit (Thermo Fisher Scientific, Waltham, MA, USA) according to the manufacturer’s instructions. Protein concentrations were determined using a protein assay kit (Bio-Rad Laboratories, Hercules, CA, USA). After electrophoresis, proteins in the gel were transferred to a nitrocellulose membrane. The membrane was then incubated at 4 °C overnight with primary antibodies against the following proteins: PCAF (Abcam), Col IV (Abcam), TGF-β1 (Santa Cruz Biotechnology, Santa Cruz, CA, USA), NF-κB (Santa Cruz Biotechnology), phospho-NF-κB p65 (Ser-536, Cell Signaling Technology, Danvers, MA, USA), Nox2 (BD Bioscience, San Jose, CA, USA), HO-1 (Thermo Fisher, Waltham, MA, USA), catalase (Abcam), SOD1 (Enzo Life Science, Inc., Farmingdale, NY, USA), NQO1 (Santa Cruz Biotechnology), Nrf2 (Santa Cruz Biotechnology), Bcl-2, Bax (all from Santa Cruz Biotechnology), β-actin (Sigma-Aldrich, St. Louis, MO, USA), and lamin B1 (Cell Signaling Technology). After washing with PBS, blots were incubated with a secondary antibody conjugated with horseradish peroxidase. Protein bands were detected by enhanced chemiluminescence reagents and imaged using Image Quant LAS 4000 (GE Healthcare, Piscataway, NJ, USA). Band densities were determined with Quantity One 1-D analysis software (Bio-Rad Laboratories).

### 4.5. Quantitative Real-Time Polymerase Chain Reaction

Total RNA was isolated from the kidney tissues using TRIzol Reagent (Thermo Fisher Scientific) according to the manufacturer’s manual. Reverse transcription was carried out to synthesize cDNA and qRT-PCR assays were performed using SYBR Premix (Takara Bio Inc., Otsu, Shiga, Japan). Primer sequences for each gene are listed in Appendix A. The specificity of the PCR product was confirmed by analyzing the melting curve. All PCRs were performed in duplicate. Results were normalized to mRNA expression in the kidney tissues of sham-operated control mice.

### 4.6. Statistical Analysis

Values were represented as the mean  ±  standard error of the mean. Statistical differences between groups were determined using one-way analysis of variance with Bonferroni correction. In all analyses, *p* < 0.05 was considered statistically significant.

## Figures and Tables

**Figure 1 ijms-20-01554-f001:**
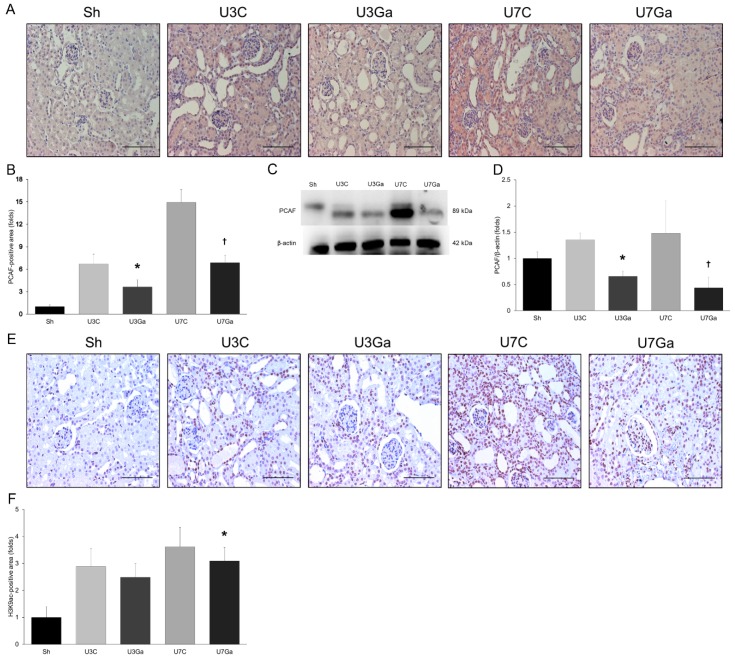
Renal expression of PCAF and H3K9ac with or without garcinol treatment after UUO. (**A**,**B**) Immunohistochemical staining and quantitative analysis showed the expression of PCAF increased in obstructed kidneys was decreased by garcinol treatment (scale bar, 2 μm). * *p* = 0.006 vs. U3C, ^†^
*p* < 0.001 vs. U7C. (**C**,**D**) Representative Western blot showed that increased renal expression of PCAF in UUO-operated mice was decreased after garcinol treatment. * *p* = 0.006 vs. U3C, ^†^
*p* < 0.001 vs. U7C. (**E**,**F**) Immunohistochemical staining for H3K9ac demonstrated that the expression of H3K9ac elevated after UUO was attenuated by garcinol treatment (scale bar, 2 μm). * *p* = 0.044 vs. U7C.

**Figure 2 ijms-20-01554-f002:**
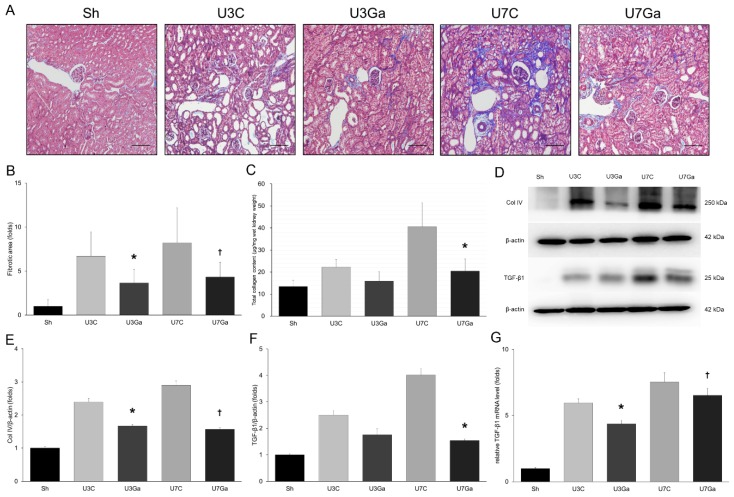
Effect of garcinol treatment on renal tubulointerstitial fibrosis after UUO. (**A**,**B**) Representative micrograph of Masson’s trichrome staining showed that marked increase in renal fibrotic lesions after UUO was attenuated by garcinol treatment (scale bar, 2 μm). * *p* < 0.001 vs. U3C, ^†^
*p* < 0.001 vs. U7C. (**C**) Renal total collagen content at day 7 post-UUO was significantly lower after garcinol treatment. * *p* = 0.001 vs. U7C. (**D**) Western blot for renal Col IV and TGF-β1. (**E**) Quantitative analysis showed that renal Col IV level was decreased after garcinol treatment. * *p* = 0.015 vs. U3C, ^†^
*P* < 0.001 vs. U7C. (**F**) Protein expression of TGF-β1 was attenuated by garcinol treatment at day 7 after UUO. * *p* < 0.001 vs. U7C. (**G**) mRNA level of TGF-β1 was decreased by garcinol treatment in UUO kidneys. * *p* = 0.001 vs. U3C, ^†^
*p* = 0.003 vs. U7C.

**Figure 3 ijms-20-01554-f003:**
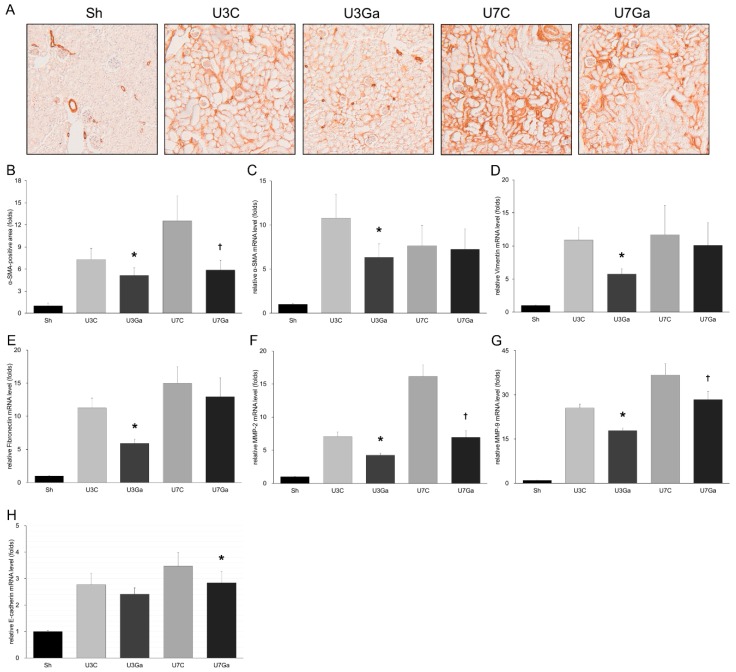
Effect of garcinol treatment on the renal expression of genes involving the epithelial-mesenchymal transition (EMT) pathway after UUO. (**A**,**B**) Immunostaining in the kidney showed that the accumulation of α-SMA after UUO was inhibited by garcinol treatment ((original magnification, ×200). * *p* = 0.002 vs. U3C, ^†^
*p* < 0.001 vs. U7C. (**C**) qRT-PCR analysis of obstructed kidney tissue showed that the renal mRNA level of α-SMA was significantly decreased by garcinol treatment at day 3 after UUO. * *p* < 0.001 vs. U3C. (**D**,**E**) mRNA levels of vimentin and fibronectin were decreased by garcinol treatment at day 3 after UUO. * *p* = 0.001 vs. U3C. (**F**) MMP-2 mRNA levels were decreased by garcinol treatment at both day 3 and day 7 after UUO. * *p* = 0.023 vs. U3C, ^†^
*p* = 0.004 vs. U7C. (**G**) MMP-9 mRNA levels were decreased by garcinol treatment at both days 3 and 7 after UUO. * *p* = 0.022 vs. U3C, ^†^
*p* = 0.001 vs. U7C. (**H**) E-cadherin mRNA level increased after UUO was significantly decreased by garcinol treatment at day 7. * *p* < 0.001 vs. U7C.

**Figure 4 ijms-20-01554-f004:**
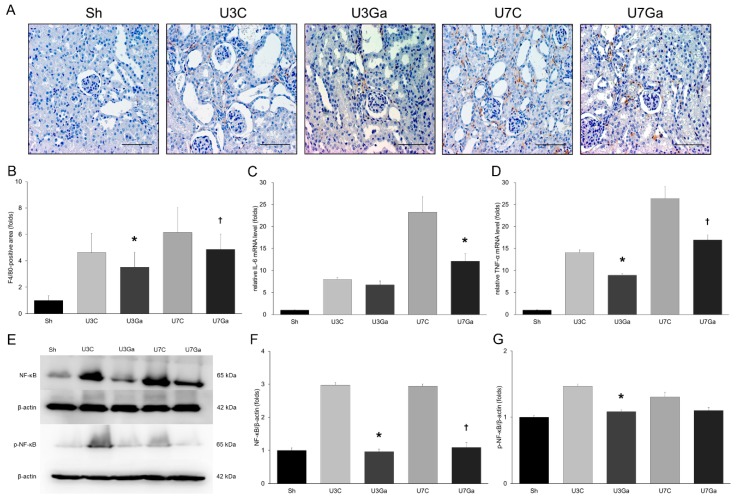
Effect of garcinol treatment on renal inflammation after UUO. (**A**,**B**) Immunohistochemistry staining showed that the number of F4/80-positive cells was increased by UUO, but decreased by the inhibition of PCAF (scale bar, 2 μm). * *p* = 0.016 vs. U3C, ^†^
*p* = 0.005 vs. U7C. (**C**) qRT-PCR analysis showed that renal IL-6 level was decreased by garcinol treatment at 7 days after UUO. * *p* = 0.029 vs. U7C. (**D**) TNF-α mRNA level in UUO kidneys was significantly attenuated by garcinol. * *p* < 0.001 vs. U3C, ^†^
*p* < 0.001 vs. U7C. (**E**) Western blot for NF-κB and p-NF-κB. (**F**) Quantitative analysis showed that renal protein expression of NF-κB was elevated by UUO, but significantly suppressed by garcinol treatment. * *p* < 0.001 vs. U3C, ^†^
*p* < 0.001 vs. U7C. (**G**) p-NF-κB expression was reduced by garcinol treatment at day 3 after UUO. * *p* = 0.001 vs. U3C.

**Figure 5 ijms-20-01554-f005:**
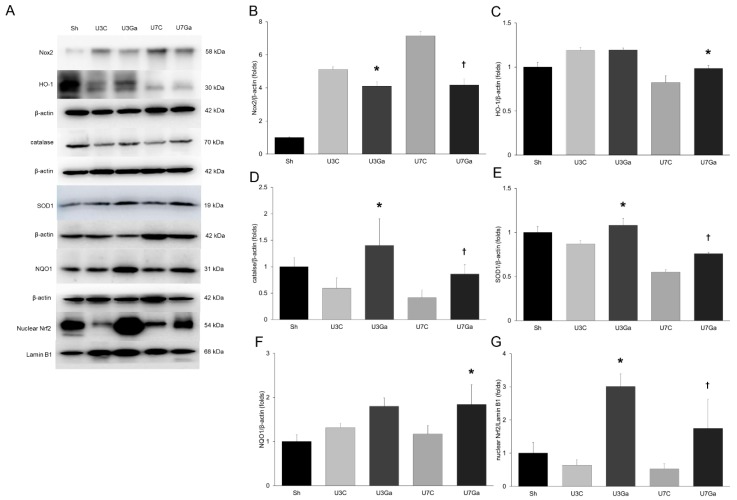
Effect of garcinol treatment on oxidative stress, antioxidant enzymes, and Nrf2 after UUO. (**A**) Western blot for Nox2, HO-1, catalase, SOD1, NQO1, and Nrf2. β-actin images were reused because the membranes for immunoblot analysis of CoI IV and TGF-β in Figure 2D were reprobed with anti-Nox2 and HO-1, and with anti-catalase, respectively. (**B**) Quantitative analysis showed that the expression of Nox2 was increased after obstructive injury, but significantly decreased by garcinol at all timepoints. * *p* = 0.015 vs. U3C, ^†^
*p* = 0.043 vs. U7C. (**C**) Renal HO-1 expression in the obstructed kidneys was restored by garcinol treatment. * *p* = 0.015 vs. U7C. (**D**) Renal catalase expression in obstructed kidneys was increased by garcinol treatment. * *p* < 0.001 vs. U3C, ^†^
*p* = 0.015 vs. U7C. (**E**) Renal SOD1 expression in obstructed kidneys was restored by garcinol. * *p* = 0.043 vs. U3C, ^†^
*p* = 0.018 vs. U7C. (**F**) Renal NQO1 expression increased in the obstructed kidneys was further increased by garcinol treatment at day 7 post-UUO. * *p* = 0.026 vs. U7C. (**G**) Obstructive injury suppressed Nrf2 expression while garcinol treatment significantly increased its expression at all timepoints. * *p* < 0.001 vs. U3C, ^†^
*p* = 0.013 vs. U7C.

**Figure 6 ijms-20-01554-f006:**
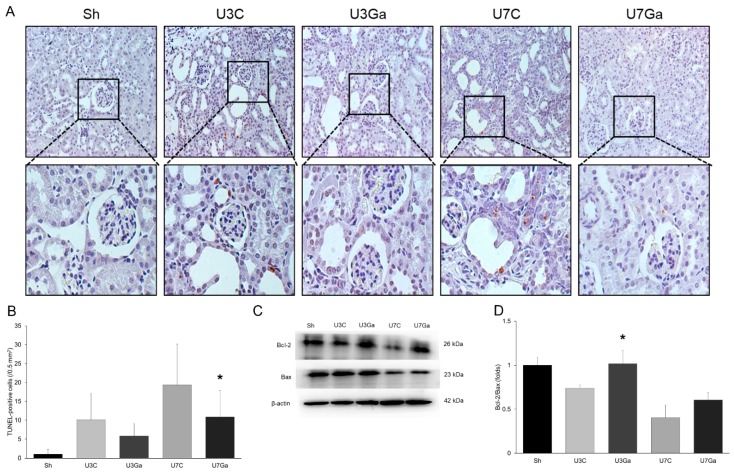
Effect of garcinol treatment on apoptosis after UUO. (**A**,**B**) Representative micrograph of TUNEL assay showed the increase of TUNEL-positive cells in the obstructed kidneys was decreased by garcinol (original magnification, ×200). * *p* < 0.001 vs. U7C. (**C**,**D**) Western blot analysis showed that the ratio of the expression of Bcl-2 to that of Bax was increased by garcinol treatment. * *p* = 0.021 vs. U3C.

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
