# Peer review of "Inhibition of p300/CBP-Associated Factor Attenuates Renal Tubulointerstitial Fibrosis through Modulation of NF-kB and Nrf2"

_ijms, 2019, doi:10.3390/ijms20071554_

Round 1
Reviewer 1 Report
The authors show a nice study; the topics is interesting and the set of experiments well described, with the rationale of each study very clear.
Few remarks:
1. The Introduction:
The description of the role of DNA acetylation and deacetylazion, before the description of our current understanding of the renal fibrosis process, is well appreciated because clarify the rationale of the study even to the readers that are not familiar with the argument.
2. Results and figures:
The panel of indicative proteins of renal fibrosis is pretty large ( α-SMA; fibronectin; vimentin etc); for each one I would show mRNA plus protein abundance.
3. The layout of FIGURE 2B should be the same as the others ( 2A, 2C,2D, 2E), for consistency.
4. The main limitations of the study are: 1) the absence of any functional information; it is clear that the inhibition of PCAF with garcinol protects from renal fibrosis in your model of kidney injury; to give a complete overview of the protective effect of garcinol, besides data on renal structure the authors could show also the effect on renal function;2) all blots and PCR have been done in total kidney, seems to me. Studying the cortex and medulla in separate experiments would provide a more complete picture. The authors should at least mention these limitations, if they cannot provide these additional information.
Author Responses1. I deeply appreciate your positive comment.
2. Thank you so much for your concern. I would like to show proteins
indicative of renal fibrosis as many as possible for thre readers.
3. I deeply appreciate your considerate indication. I corrected consistently layouts of Figure 2.
4. I deeply appreciate your careful critics. Following your pointing-out, I added following descriptions.
1) In this study, we did not measure the blood urea nitrogen (BUN) or serum creatinine as renal functional parameters. However, many previous studies have reported that BUN or serum creatinine was not significantly affected by UUO because of the presence of a contralateral kidney with good kidney function [1,2], indicating that BUN or serum creatinine is not a good indicator of renal function in an animal model of UUO [3].
1. Ning XH, Ge XF, Cui Y, An HX: Ulinastatin inhibits unilateral ureteral obstruction-induced renal interstitial fibrosis in rats via transforming growth factor β (TGF-β)/Smad signalling pathways. Int Immunopharmacol 2013, 15:406-413.
2. Wu WP, Chang CH, Chiu YT, Ku CL, Wen MC, Shu KH, Wu MJ: A reduction of unilateral ureteral obstruction-induced renal fibrosis by a therapy combining valsartan with aliskiren. Am J Physiol Renal Physiol 2010, 299:F929-F941.
3. Chung S, Yoon HE, Kim SJ, Kim SJ, Koh ES, Hong YA, Park CW, Chang YS, Shin SJ. Oleanolic acid attenuates renal fibrosis in mice with unilateral ureteral obstruction via facilitating nuclear translocation of Nrf2. Nutr Metab (Lond). 2014;11(1):2.
2) It would be the best for this study to separate renal medulla separating from renal cortex. It has been reported that glomeruli initially remain viable by remodelling of Bowman’s capsule after UUO although they could become non-functional with proximal tubular damage [1]. Cortical damage in the obstructive kidney was evidenced by a 50% reduction of the volume fraction of proximal tubular mass with unaltered relative mesangial area of glomeruli [2], suggesting that it would be justifiable to use whole kidney homogenates rather than medullary homogenates for this study.
1. Forbes MS, Thornhill BA, Minor JJ, Gordon KA, Galarreta CI, Chevalier RL. Fight-or-flight: murine unilateral ureteral obstruction causes extensive proximal tubular degeneration, collecting duct dilatation, and minimal fibrosis. Am J Physiol Renal Physiol. 2012;303:F120-F109.
2. Chaabane W, Praddaude F, Buleon M, Jaafar A, Vallet M, Rischmann P, Galarreta CI, Chevalier RL, Tack I. Renal functional decline and glomerulotubular injury are arrested but not restored by release of unilateral ureteral obstruction (UUO). Am J Physiol Renal Physiol. 2013;304:F432-F439.
Reviewer 2 Report
Chung et al. described a role p300/CBP-associated factor (PCAF) in the progression of tubulointerstitial fibrosis in a established model of renal injury. Unilateral ureteral obstruction (UUO) in C57BL/6 mice lead to tubulointerstial fibrosis detected by trichrome and α-smooth muscle actin staining as well as determination of total renal collagen content (by measuring hydroxyproline). Renal injury was associated by an increase in renal expression of PCAF and histone 3 lysine 9 acetylation. Administration of garcinol an inhibitor of the histone acetyltransferase p300/CBP-associated factor (PCAF) ameliorated these effects concurrent with an decrease in mRNA expression of transforming growth factor-β1, MMP-2, MMP-9 and fibronectin. TNF-a, IL-6 synthesis and NFkB expression as well as nuclear cytoplasmic translocation of Nrf2 was inhibited by garcinol, together with higher levels of heme oxygense-1, catalase, superoxide dismutase 1 and NAD(P)H:quinone oxidoreductase 1 compared to UUO animals. The authors conclude that PCAF may be important in the regulation of renal fibrosis in UUO by influencing inflammation, redox state and EMT. The data are interesting and relevant. Nevertheless, I have some objections: 1. Fig.2D The quantitative analysis of the expression of U3C compared to U3Ga does not reflect the representative WB presented. A quantification of the presented reproduction of the WB leads to a totally different result. Either is the quantification faulty or the blot is not representative or the reproduction is highly processed. 2. Fig. 3A To my opinion the SM-actin staining does not exhibit the expected pattern. Unfortunately, the resolution of the reproduction in the manuscript too low to judge, it appears that unspecific staining of distal tubules is presented rather than a predominantly interstitial pattern of SM-actin. 3. An inhibition of interleukin-1ß expression by garcinol isdescribed in the abstract no data are presented in the main part of the manuscript. In line 316 HK-2 are mentioned, which do not appear in the entire manuscript. This suggests, though I have no prove, that parts of the manuscript might have been recycled from other sources. minor: 4. Specify animal species of the UUO in the abstract 5. Omit horizontal stripes in Fig 2B 6. Please specify which antibodies have been utilizedand provide the concentrations applied. 7. Discussion line 214-216: The sentence is hard to understand. Do you state that activation of NF kappa B enhances its own transcription? To my opinion the wording adopted from one of the cited manuscripts is misleading. The transcriptional activity of NF kappa B is enhanced not its activation.
1. I deeply appreciate your careful examination of Fig 2D. Actually after
we performed densitometry of five to seven images of each sample, we
selected representative samples and performed western blot analysis
again. I think that we made a mistake in selecting the representative
samples or this discrepancy between densitometry and representative
image can be happened since the values of densitometry is means of all
samples. We did not absolutely process any images. For your review, I
attached the densitometry excel file.
2. I deeply appreciate your pointing-out which was very important in this
manuscript. As per your concern we performed immunohistochemistry of
SM-actin again and replaced the Fig 3A with new images. The new images
showed that it was expressed in peri-tubular area and arterioles as
well, which we think, is the typical pattern of SM-actin expression in
the kidney. Notably, in this new IHC of SM-actin, garcinol treatment
prevented UUO-induced increase of SM-actin. Therefore, the overall
conclusion of this manuscript is not changed at all.
3. I feel sorry for this stupid mistake. Although we did not show the
data, we performed analysis of IL-1 ß and used HK-2 cells, routinely in
our laboratory. At the final stage of submission, we made a decision not
to show the results of IL-1ß and those obtained in HK-2 cells because
of the bulky size of the data. I made a mistake to leave IL-1ß and HK-2
in the text, and it is true that this submission is for the first time
and was not recycled from other sources.
4. Male C57BL/6 mice were used in this study. As per your comment, we indicated the animal species in the abstract.
5. Thank you so much for your careful inspection of Fig 2B. Following your
comment, we deleted horizontal stripes in Fig 2B.
6. Thank you for your comment. We added the information of antibodies in Materials and Methods.
By acetylating histone H3 or H4, histone acetyltransferases (HATs) such as CBP/p300 and PCAF can change the chromatin structure from heterochromatin to euchromatin, which facilitates binding of transcriptional factors to promoters or enhancers in chromatin [1]. To be consistent with this hypothesis, CBP and PCAF are required for the transcriptional activity of NF-κB, the central regulator of inflammation [2,3]. Moreover, HAT activity of PCAF enhances the transcriptional activity of NF-κB [4], suggesting that PCAF may play a critical role in inflammation by modulating the transcriptional activity of NF-κB. In this study, expression of proinflammatory cytokines including IL-6 and TNF-α, transcriptional target genes of NF-κB, was upregulated associated with PCAF activation and as expected, inhibition of PCAF activity with treatment of garcinol reduced the upregulation of pro-inflammatory cytokines, resulting in prevention of renal fibrosis. Therefore, it can be hypothesized that harmful stimuli in the kidney such as UUO activates PCAF activity as well as NF-κB signaling pathway, leading to pro-inflammatory gene expression, and thus inhibition of HAT activity of PCAF contributes to anti-inflammation based on our results. Notably, garcinol treatment prevented activation of NF-κB (Figure 4) suggesting that PCAF might be involved in the activation pathway of NF-κB. Although the mechanism how PCAF is involved in activation of NF-κB in UUO-induced renal fibrosis was not investigated in this study, this finding suggests that protein kinases or molecules regulating the activity of NF-κB could be regulated by PCAF as in the case of a virulence factor of Yersinia, YopJ, acetylating IκB kinase (IKK) [5], which may highlight the feasibility of PCAF as a new therapeutic target in pro-inflammatory process.
1. Steger, DJ.; Workman, JL. Remodeling chromatin structures for transcription: what happens to the histones? Bioessays 1996, 18(11), 875-884.
2. Vanden Berghe, W.; De Bosscher, K.; Boone, E.; Plaisance, S.; Haegeman, G. The nuclear factor-kappaB engages CBP/p300 and histone acetyltransferase activity for transcriptional activation of the interleukin-6 gene promoter. J. Biol. Chem. 1999, 274(45), 32091-32098.
3. Zhao, J.; Gong, A.Y.; Zhou, R.; Liu, J.; Eischeid, A.N.; Chen, X.M. Downregulation of PCAF by miR-181a/b provides feedback regulation to TNF-α-induced transcription of proinflammatory genes in liver epithelial cells. J. Immunol. 2012, 188, 1266-1274. doi: 10.4049/jimmunol.1101976
4. Sheppard, K.A.; Rose, D.W.; Haque, Z.K.; Kurokawa, R.; McInerney, E.; Westin, S.; Thanos, D.; Rosenfeld, M.G.; Glass, C.K.; Collins, T. Transcriptional activation by NF-kappaB requires multiple coactivators. Mol Cell Biol. 1999, 19(9), 6367-6378.
5. Mittal, R.; Peak-Chew, S.Y.; McMahon,
H.T. Acetylation of MEK2 and I kappa B kinase (IKK) activation loop
residues by YopJ inhibits signaling. Proc. Natl. Acad. Sci. USA. 2006,
103(49), 18574-18579.
Round 2
Reviewer 1 Report
The majority of comments and suggestions have been addressed.
No more concerns.
Reviewer 2 Report
In their revised manuscript the authors try to elucidate the role of PCAF in the regulation of renal fibrosis in C57BL/6 mice with unilateral ureteral obstruction.
They have adequately addressed all my concerns and have significantly improved the quality of the manuscript.
I have not further major objections.
Before publication the language appears to need some improvement, especially in the introduction section.
Additionally, it would be helpful to increase the size of the reproductions of the immunohistochemical stainings to facilitate a better evaluation of their significance.